# Convergent evolution of fern nectaries facilitated independent recruitment of ant-bodyguards from flowering plants

Jacob S. Suissa [1] ✉, Fay-Wei Li [2,3] & Corrie S. Moreau [4,5]

Plant–herbivore interactions reciprocally influence species' evolutionary trajectories. These interactions have led to many physical and chemical defenses across the plant kingdom. Some plants have even evolved indirect defense strategies to outsource their protection to ant bodyguards by bribing them with a sugary reward (nectar). Identifying the evolutionary processes underpinning these indirect defenses provide insight into the evolution of plant-animal interactions. Using a cross-kingdom, phylogenetic approach, we examined the convergent evolution of ant-guarding nectaries across ferns and flowering plants. Here, we discover that nectaries originated in ferns and flowering plants concurrently during the Cretaceous, coinciding with the rise of plant associations in ants. While nectaries in flowering plants evolved steadily through time, ferns showed a pronounced lag of nearly 100 My between their origin and subsequent diversification in the Cenozoic. Importantly, we find that as ferns transitioned from the forest floor into the canopy, they secondarily recruited ant bodyguards from existing ant-angiosperm relationships.

Coevolutionary theory examines the reciprocal influences between interacting species[1]. Whether predator-prey dynamics or mutualistic relationships, these interactions can shape a lineage's evolutionary trajectory[2,3]. Classic coevolutionary models predict that, in certain cases, predators and their prey can be entangled in an evolutionary arms race[4,5]. Reciprocal and strong survival pressures exert their influence on prey, prompting the development of defensive strategies against predators, and on predators, driving the evolution of traits to consume prey. Given the unique dynamics of sessile plants and their mobile predators, plant predation (herbivory) has led to many defense syndromes across the plant kingdom[6,7]. For instance, the milkweeds and their relatives produce a pharmacy of chemical compounds ready to be deployed upon damage[8]. Other species have evolved physical defenses to deter would-be herbivores, including spines in cactus, stinging hairs in nettles, and latex in figs.[9–11]. Interestingly, certain lineages like the iconic Ant Acacia have outsourced their anti-herbivore

strategy by enlisting an external defensive force of ants[12]. The evolution of these indirect defense traits is of particular interest as they bridge the gap between predation and mutualism, inexorably linking species across kingdoms of life. Identifying the processes underpinning these indirect defense strategies is paramount for understanding how mutualism can shape the diversity of complex traits.

To engage in these types of anti-herbivore strategies, plants must bribe defensive insects (usually ants) with rewards. These rewards include domatia (e.g., plant-made structure where symbiotic ants or other organisms nest), Beltian bodies (protein or lipid-rich deposits[13]), or in most cases nectar (sugary liquid) secreted from a structure called a nectary. Unlike floral nectaries, which aid in pollination, nectaries involved in defense commonly occur on leaves and are called extrafloral nectaries (EFNs)[12]. By attracting ants, EFNs are an effective mechanism to ward off arthropod herbivores[14,15]—in some cases reducing foliar herbivory threefold[16]. Extrafloral nectaries are

[1]Department of Ecology and Evolutionary Biology, University of Tennessee Knoxville, Knoxville, TN, USA. [2]Boyce Thompson Institute, Ithaca, NY, USA. [3]Plant Biology Section, School of Integrative Plant Science, Cornell University, Ithaca, NY, USA. [4]Department of Ecology and Evolutionary Biology Cornell University, Ithaca, NY, USA. [5]Department of Entomology, Cornell University, Ithaca, NY, USA. ✉e-mail: jsuissa@utk.edu

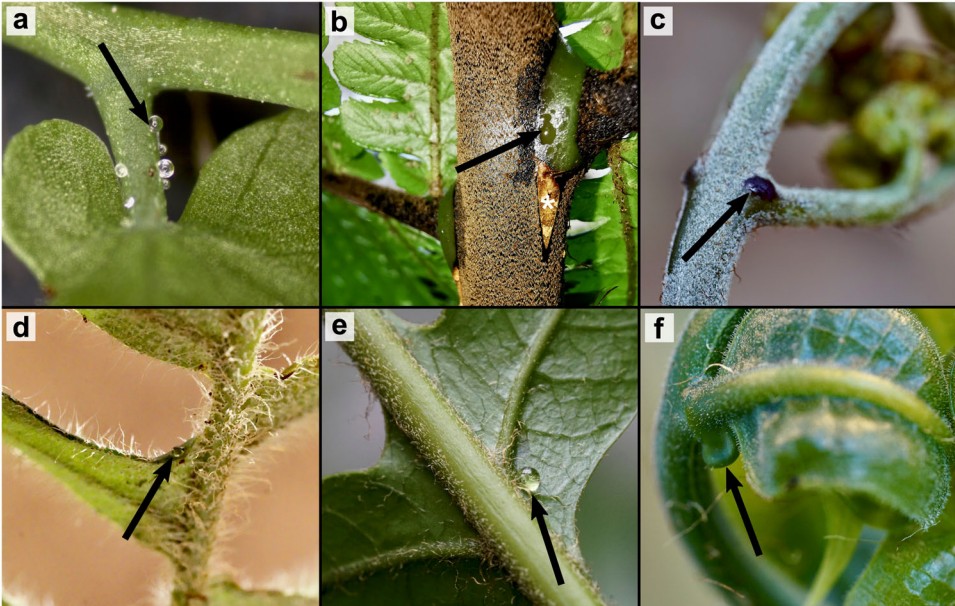

**Fig. 1 | Diversity of fern nectaries. a** gland-tipped nectar-secreting trichomes of *Lygodium microphyllum*. **b** raised nectar gland of *Gymnosphaera henryi*. **c** raised pigmented nectar gland of *Pteridium aquilinum* **d** microscopic adaxial nectar pore of *Pleopeltis thysannolepis*. **e** microscopic abaxial nectar pore of *Drynaria pilosa*. **f** cup-like nectary of *Drynaria speciosa*. Photographs from JSS, except *G. henryi* which was provided with permission by Shiyong Dong.

widespread and documented in nearly 4000 flowering plant species (Supplementary Data 1[17]). Interestingly, nectaries also occur in ferns (Fig. 1a–f[18])–an old observation made by Charles Darwin and his son Francis Darwin[19]. Since their initial discovery, research on fern nectaries has mainly explored their presence, structure, and ecology[20–23]. While not as variable or abundant as in flowering plants, fern nectaries are still quite diverse (Fig. 1). Many are inconspicuous and produced at the margins of the lamina or vein junctions such as in the Bracken fern (*Pteridium aquilinum*; Fig. 1a–e). However, some species like *Drynaria speciosa* can produce highly modified cup-like structures (Fig. 1f), similar to certain EFNs in angiosperms like cherries (*Prunus* spp[24,25].). While seldom explored, nectar composition in ferns may be similar to that of angiosperm EFNs, containing simple and complex sugars as well as amino acids, peptides, and other macromolecules[18,26]. While there is some uncertainty on the role of nectaries in ferns (especially temperate species[27–29]), ecologically, fern nectaries can also be implicated in indirect ant-defense[30]–with the artificial occlusion of nectaries significantly increasing herbivory and larval herbivore presence compared to neighboring unblocked leaves[23,31]. While the relationships between ferns and ants has been acknowledged for over a century[32–35], the origin and timing of these interactions–as well as the convergent evolution of ant-EFN mutualism in angiosperms–has remained unknown.

In this work, we explore the convergent evolution of nectaries and ant-mediated herbivore-defense in ferns and flowering plants. We assembled a cross-kingdom dataset, amalgamating nectary presence data for 149 ferns and 3913 angiosperms, plant association information in 1341 ants, and fern-feeding preference for 725 arthropod herbivore lineages. We integrated this dataset with phylogenetic comparative methods to explore the evolutionary origin, diversification, and potential drivers of nectary evolution. We first implemented stochastic character mapping to delve into the evolution of nectaries across ferns and flowering plants as well as plant associations in ants. Using these data, we estimated the timing of key evolutionary events, including nectary origin in ferns and flowering plants, the initiation of plant associations in ants, and the emergence of fern-feeding arthropod herbivores. We then implemented hidden Markov models to assess correlated trait evolution and used hidden state speciation and

extinction models to estimate the impact of nectary presence on lineage diversification in ferns. Our analyses uncovered major insights on the convergent evolution of nectaries and the role of biotic interactions and herbivory in ferns. While the convergence of nectaries in ferns and flowering plants has been known for over a century, we found that the timing of nectary evolution in both lineages mirror each other, with potential origins in the Cretaceous. We further observed that these Cretaceous origins of nectaries in both lineages corresponded with, but slightly predate, the rise of plant associations in ants. However, the diversification of ferns with nectaries shows a nearly 100 My delay from their origin; this lag may relate to the rise of fern-feeding arthropod herbivores in the Cenozoic, although the evidence supporting this hypothesis is presently limited. Our analyses further uncover a link between fern nectary evolution and changes in growth habit, suggesting that ferns secondarily recruited ant bodyguards from existing ant-angiosperm interactions during the fern escape from the forest floor into the canopy. These insights suggest that similar evolutionary and ecological dynamics underpin the convergent evolution of ant-plant mutualisms across clades and macroevolutionary scales.

## Results and discussion
### Cretaceous evolution of nectaries in ferns and angiosperms
Ferns and flowering plants are the only two lineages that have evolved nectaries (barring hypotheses on the pollination droplet in Gymnosperms[36,37]). This is a remarkable example of convergent evolution, especially considering that ferns and flowering plants diverged from their most recent common ancestor more than 400 million years ago[38,39]. However, until now, we lacked a detailed understanding of the timing, scale, and lability of nectary evolution within each respective clade. Among ferns, we found three major ancestral origins at the base of the Cyatheaceae, Polypodiaceae, and Lygodiaceae (we also found independent origins of nectaries in *Pteridium*, Dennstaedtiaceae and *Polybotrya*, Dryopteridaceae; Fig. 2, Supplementary Figs. 1–3). These ancient origins were followed by multiple instances of nectary loss and subsequent gain. The notion of an ancient singular origin across each lineage is interesting, especially considering the diversity of nectaries observed within each family and across ferns. These observations are similar to those of EFNs in the flowering plant genus *Senna*[40], which is

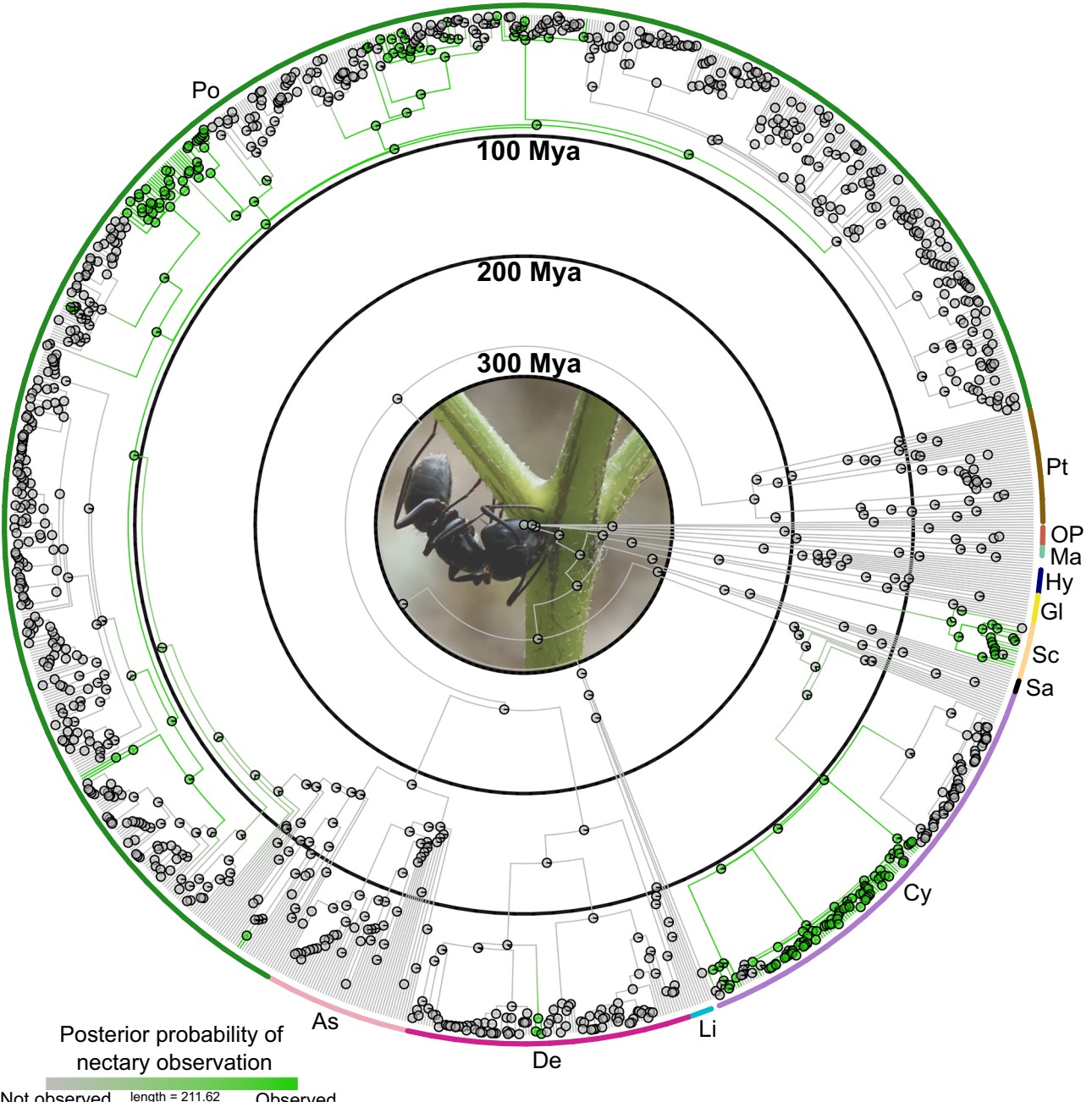

**Fig. 2 | Ancestral character estimation of nectary evolution in ferns.** Genera that lacked nectaries were pruned to 1 tip per genus. Ancestral character estimation suggests that nectaries had the potential to evolve in the Cretaceous. However, while nectaries likely had a deep origin, many of the lineages with nectaries did not diversify until the Cenozoic. Ancestral character estimation was implemented using stochastic character mapping. Pie charts at nodes along the time-calibrated fern phylogeny represent ancestral states calculated as the marginal posterior probability of nectary presence (green) or unobserved (gray). Colors along the branches represent estimated character states summarized across 100 randomly selected posterior samples of character histories from 1000 stochastic character maps. Bars at tips indicate the species character state for nectary presence or absence. Bars along the perimeter of the reconstructed tree and bolded letters indicate major fern clades. Light pink: Aspleniineae, Green Polypodiineae, Brown: Pteridineae, Red: Equisetales, Dark orange: Ophioglossales & Psilotales, Teal: Marattiales, Blue: Hymenophyllales, Yellow: Gleicheniales, Cream: Schizaeales, Black: Salviniales, Purple: Cyatheales, Light blue Lindsaeineae, Pink: Dennstaedtiineae. Black inlaid circles indicate 100-million-year time intervals. Photo inset of an ant feeding on fern nectar taken by JSS.

also hypothesized to have a single ancestral origin with many losses and positional modifications across their evolutionary history. An ancient origin across each major lineage could reflect an ancestral genetic or developmental pathway that has been co-opted across multiple lineages; however, this is unlikely given the diversity of structure (Fig. 1) and hypothesized homology of nectaries across ferns (Supplementary Notes). Alternatively, it could reflect an ecological or evolutionary predisposition to evolving nectaries—indeed, a hypothesis proposed to explain the phylogenetic patterns of EFNs across the whole bean family, Fabaceae[41]. This early estimated predisposition could even represent an unrelated chance event, such as ants feeding on sap from plant wounds[3,42,43].

We further established that the origin of nectary development in ferns occurred during the Cretaceous, with an average potential origin

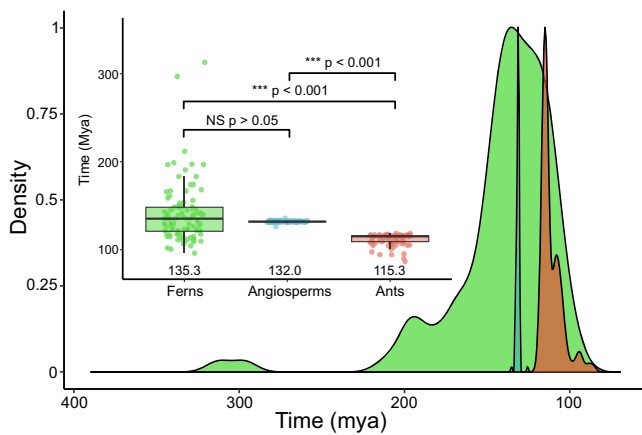

**Fig. 3 | Distribution of timing around nectary origin in ferns (green), EFNs in flowering plants (blue), and plant associations in ants (red), derived from 100 stochastic character maps.** Boxplot inset depicts a set of unpaired two-sided Wilcox tests between the timing of each trait with a Bonferroni correction. The center line within each box denotes the median value, while the bounds of the box represent the interquartile range (IQR), encapsulating the middle 50% of the data, the lower portion of the box represents the 25th percentile, while the upper portion of the box denotes the 75th percentile. The whiskers extend from the box to the minima and maxima, respectively. The timing of nectary origin and ferns and flowering plants is not significantly different, while ant-plant associations seemed to have evolved slightly later. *P*-value for each pairwise comparison is as follows: Ferns:Angiosperms ($p = 0.35$), Ferns:Ants ($p < 2.22e^{-16}$), Angiosperms:Ants ($p < 2.22e^{-16}$).

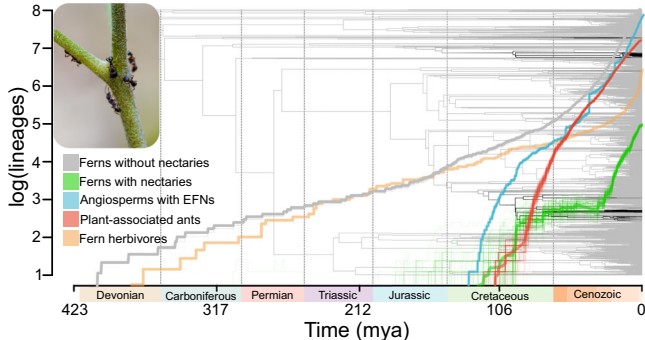

**Fig. 4 | Cross-kingdom lineage-through time plot.** Nectaries across ferns, EFNs in flowering plants, and ants associated with plants likely originated around the middle Cretaceous. However, an almost 100-million-year lag in the diversification of ferns with nectaries may relate to the rise of fern insect herbivores in the Cenozoic (however, we have limited support for this hypothesis). Lineage-through-time (LTT) plots from 100 stochastic character maps for ferns with nectaries, angiosperms with extrafloral nectaries, and ants associated with plants. Each line represents a single LTT plot for 1/100 stochastic character maps, the thicker central lines represent the average LTT across all maps. Fern herbivore LTT plot is derived from a singular tree given the data used. The LTT plot shows the cumulative number of lineages through time for each group, as indicated by the colored lines on the graph. The x-axis represents time in millions of years before the present (mya). Inlaid in the plot is a stochastic character map indicating the ancestral state reconstruction of the presence or absence of fern nectaries (for more detail see Fig. 2). Black lines are fern lineages hypothesized to possess nectaries and those in gray are lineages unobserved nectaries. Photo inset of ants feeding on Bracken fern nectaries taken by JSS.

of 135.3 mya, ±33.6 mya (Mean: 135.10 mya; Min: 96.20 mya; Max: 312.90 mya), very late in fern evolutionary history. Interestingly, our full observed time range of nectary origin in ferns encompasses previous hypotheses suggesting a Jurassic origin, around 200 mya[44]. However, our large dataset enables us to refine these hypotheses, pinpointing a more specific origin approximately 135 mya. This contrast with earlier hypotheses emphasizes the previous lack of supporting data, which precluded a consensus on the timing of fern nectary origin. Our comprehensive dataset provides statistical evidence, enabling us to offer a clearer understanding of nectary origin in ferns.

Our recovered temporal patterns of nectary origin in ferns closely parallel the emergence of EFNs in angiosperms (Median: 132.0 mya Mean: 132.00 mya; Min: 126.20 mya; Max: 135.90 mya; SD: 1.00 mya), with no significant distinction in the estimated age of origin between both clades (Figs. 3, 4). To our knowledge, there is no fossil evidence of nectaries in ferns and the oldest evidence of EFNs in flowering plants is from a *Poplar* that dates back to the Oligocene[45]. It is possible that nectaries could have evolved much earlier in ferns. This is reflected in a probability of nectary origin in Schizaeales (Fig. 2). Moreover, while extant Marattiales do not have nectaries, they do have potential ant associations (as it has been observed that they can live within the stipules; A. G. Murdock pers. comm.), this could hint at an ancient relationship. Whether the timing of nectary origin that we uncovered survives the scrutiny of future analyses or new fossil evidence, our current estimates imply that nectaries evolved very late in the ferns, yet quite early in the evolutionary history of angiosperms.

In addition to convergence in origin, the degree of lability in nectary evolution is also striking across ferns and flowering plants. In ferns, we observe nectaries in 149 species, across 12 genera, 5 families, and 3 orders (Fig. 2; Table 1). In contrast to the ferns, EFNs occur in a total of 3913 flowering plant species spanning 796 genera, 119 families, and 42 orders (Supplementary Fig. 5)—numbers certain to change with additional survey. While the overall number of species with nectaries is 26-times larger in flowering plants compared to ferns, the relative

proportion of species is nearly identical. In both lineages nectary-bearing species account for 1% of total species diversity, 6% of generic diversity, and 23–26% of familial diversity (Table 1). The only exception to this is that 65% of angiosperm orders have EFNs compared to only 43% of fern orders. The phylogenetic lability of nectary evolution in both clades suggests that they may be relatively 'easy' to evolve and modify across lineages (Fig. 2; Supplementary Figs. 1, 3, 5[17,30,46,47]). This is also evident morphologically. For instance, EFNs in the angiosperm genus *Viburnum* occur in variable positions on the leaf, including at the base of the lamina, petiole, or leaf margins. While they evolved only a single time in the genus, their number and position on the leaf changed drastically over evolutionary time[48]. Likewise, in the basket ferns *Drynaria* spp., we observed that nectaries have a single ancestral origin (Fig. 2) but are positionally variable, either occurring near vein junctions (Fig. 1e) or on highly modified cup-like structures (Fig. 1f), also observed by others[18,25]. Nectaries pose a unique indirect defense trait where multiple factors may underly their lability. While of course, the numerous gains could be explained by increased herbivore defense, the high number of losses could relate to selection against freely distributing sugar (e.g., due to increased microbial activity on nectar[49]), or (a loss of potentially valuable carbon). While there are several hypotheses for the ultimate explanation of nectary origin[50,51], their lability is certainly impressive across broad and narrow time scales. Our phylogenetic and morphological observations suggest that similar evolutionary and ecological dynamics may occur across ferns and flowering plants despite their over 400 million years of evolutionary divergence.

## Origin of nectaries in ferns and angiosperms coincide with the rise of plant-associated ants

The diversification of major insect families and the increases in arthropod-plant relationships also occurred in the Cretaceous[52,53]. One of these insect lineages were the ants, which originated nearly 150 mya[42,54,55]. While they were ancestrally terrestrial carnivores[56–58], ants transitioned multiple times to herbivory and the arboreal habit in

**Table 1 | Diversity of species, genera, families, and orders of ferns with nectaries and flowering plants with EFNs in this study**

| Clade | Species | Genera | Families | Orders | Gains | Losses |
|---|---|---|---|---|---|---|
| Ferns | 149 (1%) | 12 (6%) | 5 (23%) | 3 (43%) | 17 (20%) | 68 (80%) |
| Angiosperms | 3913 (1%) | 796 (6%) | 119 (26%) | 42 (65%) | 1555 (40%) | 2318 (59%) |

Columns Species, Genera, Families, and Orders denote the number of the respective taxonomic class with nectaries. Percent of the total observed diversity at each taxonomic level recorded in paratheses. Gains and losses denote the average raw number of gains and losses and the percent of total state changes across the species-level phylogenies.

the Cretaceous, nearly 115.3 mya (Mean: 111.8 mya, Min: 86.9 mya, Max: 118.7 mya, SD: 6.3 mya; Figs. 3, 4; Supplementary Fig. 6). The transition from carnivory to herbivory may not have been that large of a physiological or metabolic leap. Indeed, some of the most iconic ant carnivores feed on plant products in some capacity. For instance, *Odontomachus* and *Paraponera*—ants with putative adaptations to carnivory such as large jaws, stingers, and aggressive behaviors— commonly ingest plant material[59–61]. Importantly, many extant ants have been observed to feed on sugar-rich sap exuded from plant wounds across a diverse array of tree species, potentially spurring the origin of these ant-angiosperm mutualisms[43].

The gain of plant associations in ants seems to slightly lag behind the origin of EFNs in flowering plants. It is possible that this may relate to the observation that ants are not the only insects to feed on extra-floral nectar. Indeed, in high-elevation areas where the flowering plant genus *Inga* can grow, nectivorous ants may be absent, and predatory or parasitoid wasps (lineages which predate the origin of ants) have been observed to feed on extra floral nectar[30,62]. However, once ants start to evolve plant relationships, the diversification patterns of plant-associated ants and angiosperms with EFNs seems co-linear and tends to increase steadily (Figs. 2, 4, Supplementary Fig. 6; Supplementary Data 4). This potential link between ants and flowering plants has been demonstrated to coincide not just with EFNs but also other ant-attracting structures like elaiosomes and domatia[42,63]. In contrast, while nectaries in ferns likely had the potential to evolve at the same time as EFNs in angiosperms and plant-associated ants, the majority of nectary-bearing fern lineages did not diversify until the Cenozoic (Fig. 4). The link between flowering plants and ants may be stronger than with ferns, because EFNs may also serve a dual function by 'distracting' ants from visiting floral nectar—which may negatively impact reproduction[64,65]. Interestingly, soral nectaries have not been identified in ferns, but this trait may exist would potentially facilitate zoochorous spore dispersal. Regardless, from our observed patterns and the modern prevailing associations of ants and angiosperms, we hypothesize that ant-plant mutualism initiated in the flowering plants and most ferns secondarily tapped into this established relationship despite their longer evolutionary history.

Providing support for this hypothesis is the observation that nectary-bearing ferns are disproportionately enriched in canopy-dwelling lineages, including the tree ferns Cyatheaceae, the climbing ferns (*Lygodium*), and select genera in the dominant epiphytic Polypodiaceae (*Pleopeltis, Campyloneuron, Niphidium, Drynaria, Platycerium, Selliguea*, and *Serpocaulon*; Fig. 5). Since these lineages mostly occur in the canopy or sub-canopy (i.e., epiphytes, climbers, or tree ferns) we describe them as canopy-dwelling. We find a significant phylogenetic correlation between growth habit (canopy vs. understory) and nectary presence (Supplementary Data 1). Our best-fitting model also supports rate heterogeneity in across the fern phylogeny. We found two rate categories (a fast rate and a slow rate) best categorize the correlated evolution of nectary presence and growth habit. While only a few total transitions occur at all, nectary gains are 3-times more likely in canopy-dwelling lineages (Supplementary Data 2). Further, our model recovered no transition from the canopy, nectary-bearing taxa to the understory, nectary-bearing taxa. This could be driven by the observation that most ferns are epiphytic; however, such

a stark and clear pattern suggests an ecological link between the canopy-dwelling habit and nectary presence. This association of epiphytism and ant association is also observed in other traits, for instance, *Lecanopteris* (which interestingly lacks nectaries) is genus of ferns that produce ant-using domatia and the genus is entirely epiphytic[66]. While it's worth noting that many nectary-bearing ferns are epiphytic[67], our observed patterns may not be solely attributed to epiphytism. This is underscored by the presence of nectaries in many tree ferns, suggesting that factors beyond epiphytism could also be influencing these patterns.

The underlying reason why nectary-bearing ferns are over-represented in canopy-dwelling species may be that epiphytic and climbing ferns grow in angiosperm canopies, providing a nearby habitat for arboreally nesting ants. Similarly, tree ferns provide a comparable niche to the angiosperm canopy. The proximity of epiphytes, climbers, and tree ferns to angiosperm canopies likely facilitated the transition of angiosperm-associated ants onto ferns. These observations suggests that the ancient fern lineage adopted the mutualistic defense strategy of attracting ants through nectaries as they migrated up and off the forest floor.

### Cenozoic diversification of nectary-bearing ferns

Evolutionary lags are common patterns observed in mutualism[68]. For instance, temporal lags in co-diversification have been observed between angiosperms and herbivorous weevils[69]. Here, we observed a nearly 100-million-year lag between the Cretaceous origin of nectaries in ferns and the accumulation of lineages bearing nectaries in the Cenozoic (Fig. 4). This observed lag could be due to phylogenetic error as diversification patterns are directly tied to the completeness of the phylogeny. While the phylogeny we used here is the most comprehensive to date, it does lack 60% of fern species diversity[39]. These patterns could also arise through the background diversification of all ferns during this time. However, the diversification patterns of ferns with nectaries appears to deviate from patterns of ferns without nectaries, as non-nectary-bearing ferns have a consistent trend over time with a slight acceleration near the beginning of the Cretaceous (Fig. 4).

An alternative hypothesis may be that the diversification of nectary-bearing ferns in the Cenozoic was driven by the rise of fern-feeding arthropod herbivores. Indeed, Ehrlich and Raven hypothesized that herbivores can drive the diversification of plant groups[4]. While the underlying cladogenic mechanism has been debated[70,71], the interplay between herbivory, plant defense, and reproductive barriers likely facilitate herbivore-mediated plant speciation. While focused on angiosperms, these complex herbivore-mediated diversification patterns have been observed in other groups such as in the bryophytes and their phytophagous Dipteran herbivores[72]. To explore the evolutionary correlation between fern-feeding arthropod herbivores and nectary-bearing ferns, we asked whether these lineages have similar diversification patterns. The best-fitting diversification model for ferns with nectaries suggests that they experienced a 6-fold increase in diversification during the Cenozoic (Supplementary Data 4). While we uncover shifts in fern herbivore diversification rates and an accumulation of lineages in the Cenozoic (Fig. 4; Supplementary Figs. 7, 8, 10), these patterns are not statistically supported (Supplementary Data 3).

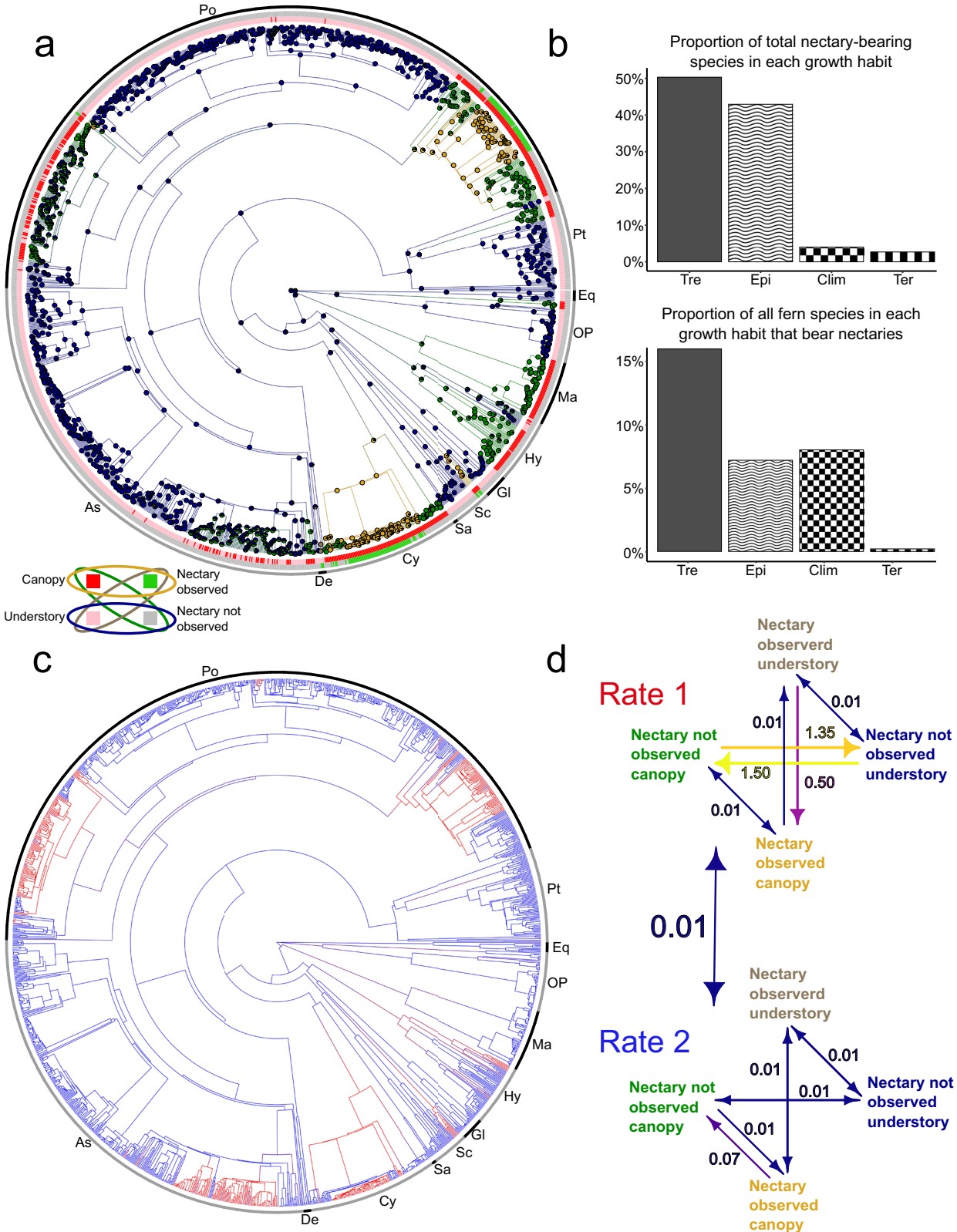

It is possible that this observation suggests that nectaries do not provide sufficient anti-herbivore benefit to the ferns. Indeed, many ferns do have complex anti-herbivore biochemistry[73], and some ants observed to feed on fern nectaries are not aggressive[27].

Moreover, there may be contrasting interactions between particular types of insect herbivores such as chewing and sap-sucking insects. Sap-suckers may attract ants to 'protect' and 'tend' them,

leading to direct competition between insect-nectar and plant-nectar[74]. A more detailed analysis of herbivore groups and the complex multi-species interactions may resolve these fine-scale relationships. Interestingly, however, our observations of increased nectary presence in ferns corresponds with a more general pattern of increased Cenozoic herbivory identified in the fossil record[75,76]. Undoubtedly more information is needed to confidently determine

**Fig. 5 | Evolution of nectaries in ferns facilitated by shifts from understory to canopy-dwelling habitat. a** Summary of 50 sampled stochastic character maps showing the transitions and correlated evolution between growth habit and nectary presence (colors indicating dual states shown in the inset in the bottom left of plate **a**). Ancestral character estimation was implemented using stochastic character mapping. Pie charts at nodes along the time-calibrated fern phylogeny represent ancestral states calculated as the marginal posterior probability of nectary presence and growth habit. Blue: nectary not observed understory, Green: nectary not observed canopy-dwelling, Yellow: nectary observed canopy-dwelling, Brown: nectary observed understory. Colors along the branches represent estimated character states summarized across 50 randomly selected posterior samples of character histories from 100 stochastic character maps. Bars at the tips indicate the species character state for the dual trait. **b** Distribution of fern species with nectaries and their growth habit. Top: Proportion of total nectary-bearing species in each growth habit. Of the 149 fern species bearing nectaries most were tree ferns (Arb: 50.3%), then epiphytic (Epi: 43.0%) and climbing (Climb: 4.0%), with the fewest species being understory (Terr: 2.7%). Bottom: Proportion of total fern species in

each growth habit-bearing nectaries. Nectary-bearing species are disproportionately enriched in canopy-dwelling species (Arb: 15.33%), (Epi: 6.90%), (Clim: 7.6%), (Ter: 0.19%). **c** Phylogeny with red highlighted branches depicting which parts of the phylogeny are in Rate 1 (red) or Rate 2 (blue). Colors along the branches represent estimated rate category summarized across 50 randomly selected posterior samples from the stochastic character maps. **d** Best-fitting model from CorHMM depicting the transition rates between dual characters. Rates are individual changes per million years. Colors of arrows related to the speed of transition (cool being lower and hot being faster). Transition between rate categories denoted by large blue arrow in between both matrixes. Bars along the perimeter of the reconstructed trees and bolded letters indicate major fern clades. As: Aspleniineae, Po: Polypodiineae, Pt: Pteridineae, Eq: Equisetales, OP: Ophioglossales & Psilotales, Ma: Marattiales, Hy: Hymenophyllales, Gl: Gleicheniales, Sc: Schizaeales, Sa: Salviniales, Cy: Cyatheales, De: Dennstaedtiineae. Canopy-dwelling indicates elevated leaves (i.e., Tre: tree ferns, Epi: epiphytic, Clim: climbing). Understory, non-canopy-dwelling species denoted as Ter (terrestrial).

what drove the Cenozoic increase of nectary-bearing ferns; however, our observations hint at a possibility that increased herbivory may have contributed.

As has been hypothesized across other lineages[3], extrafloral nectaries may promote lineage diversification across ferns. The underlying mechanisms of nectary-mediated speciation are not fully clear, but the presence of ant bodyguards may enable plant species to expand into new ecological niches without the threat of excessive predation[77]. As most ferns with nectaries occupy canopy-dwelling niches (tree ferns, epiphytes, climbers; Fig. 5b) and there are higher transition rates to canopy-dwelling habits when nectaries are present (Fig. 5d), this is an enticing hypothesis (Fig. 5b). However, our diversification rate analyses suggest that, on average, fern lineages with nectaries are not diversifying faster than non-nectary-bearing lineages (Supplementary Figs. 11–14; Supplementary Data 5). If nectaries are important in the diversification of some fern lineages, it is possible that an insufficient amount of time has passed for these nectary-bearing lineages to accumulate more species and reflect phylogenetic patterns—similar to non-equilibrium dynamics in floral evolution across angiosperms[78]. While protection from herbivory is undoubtedly beneficial for individual survival, nectary-driven mutualism is just one mechanism of herbivore defense. Other ferns likely invest in secondary metabolites[73] or physical defenses[79], which may be one reason we do not see an association between nectary evolution and lineage diversification.

In summary, our results provide insights on the role of plant-animal interactions in shaping fern and angiosperm evolutionary history. We demonstrate that ant-attracting nectaries occur in a similar proportion of total species diversity across ferns and flowering plants and that nectary evolution is labile across both clades. Our analyses suggest that nectaries had the potential to originate around the same time in ferns and flowering plants, and this timing corresponds with the rise of ant-plant associations in the Cretaceous. Given the lagged diversification of nectary-bearing ferns and the correlation between nectary presence and shifts away from the forest floor into the canopy, it is likely that ferns independently recruited ant defenders by tapping into established ant-angiosperm interactions. This suggests that the ancient fern lineage adopted this mutualistic defense strategy after flowering plants, despite the longer evolutionary history of ferns. While assessments of rates through time across lineages has been suggested to be compromised by statistical artifacts[80], lines of additional evidence including ecological and observational data further support our interpretations. The similarity in convergence and evolution of nectaries across ferns and flowering plants may suggest that shared rules of life  govern the evolution of ant-mediated defense across evolutionary and ecological scales.

## Methods
### Data collection
Four datasets were collected for this study. The first dataset included on the presence of nectaries in ferns and was generated using the literature records[18], consulting with experts, and observing herbarium specimens (Supplementary Data 5). Study species were chosen based on their presence in the Fern Tree of Life project (FTOL[39]). Relevant publications on the study species were identified through a comprehensive search of the scientific literature using keyword searches, citation tracking, and consultation of reference books and field guides. We corroborated trait data and generated additional observation data using herbarium specimens selected from the Herbarium of the L. H. Bailey Hortorium (BH) at Cornell University. We further gathered information on the growth habit of ferns bearing nectaries using previously published data from[81].

The second dataset comprised the presence of extrafloral nectaries in angiosperms and was obtained from Weber et al.[17] and the World List of Plants with Extrafloral Nectaries[82]. For simplicity the term nectaries in this manuscript is used to describe vegetative nectaries, that is extrafloral nectaries in flowering plants and any nectary structure in ferns (since ferns lack flowers the term extrafloral is superfluous). Species names in the dataset were manually corrected and synonyms updated based on the presence of taxa in the Smith and Brown 2018 Angiosperm Phylogeny[83]. We used the R program Taxize to obtain families and orders for species in the phylogeny[84]. In total, there were 353,185 tips in the angiosperm phylogeny. Since certain phylogenetic comparative methods such as *make.simmap* in the phytools package[85] may not work on large topologies, we reduced the size of the dataset for downstream analysis. Each genus that lacked nectaries was randomly pruned to include only 1 taxon; for genera that had nectaries, we kept all tips in the genus. In total, this dropped 286,707 tips from the tree. We then pruned the dataset further by randomly pruning the total number of species without nectaries by 90%. Finally, we dropped all gymnosperms from the tree, and resolved any polytomies using the *multi2di* function in ape[86], and made the tree ultrametric using *force.ultrametric* in Phytools[85]. After filtering, the tree had a total of 9589 tips including 2636 tips with EFNs.

The third dataset contained information on ant diversification and host interactions, which was gathered from Nelsen et al.[42]. We filtered the data based on ant-plant relationships, retaining only data on ants that nest in, eat, or interact with plants (excluding leaf cutters). The original dataset from Nelsen et al.[42] included all ant lineages (n = 1731). We coded specimens by the presence or absence of plant associations. Ants were considered to have plant associations if they had any plants in their diet, foraged in a canopy, or nested in a canopy. This included ants that were plant-specialists and generalists as both are known to

visit nectaries. After filtering we were left with a total of 1341 ant species.

Our fourth dataset comprised information on known fern-feeding arthropod herbivores and included data on 809 species from Fuentes-Jacques et al.[87]. While this dataset is robust it is not comprehensive. There are likely many thousands of more arthropod species that feed on ferns. Building a comprehensive phylogeny of all the arthropod species that feed on ferns is currently not feasible in this study due to the enormous number of clades involved and the lack of a well-established arthropod phylogeny. Therefore, to address this challenge, we used an indirect phylogeny-inference approach, TimeTree5[88]. TimeTree5 is a tool designed to build chronograms (time-calibrated phylogenies) using a database of published studies, integrating phylogenetic trees and molecular data to estimate evolutionary timelines. We uploaded a list of all genera identified to feed on ferns to TimeTree5[88]. The program compiled and built a chronogram with all of the representative species in each genus (or related genus) of fern herbivores, which had publicly available data. This allowed us to estimate the diversification patterns and timing of major groups of fern-feeding arthropod herbivore, despite the lack of comprehensive phylogenetic data on all the arthropod fern herbivores. While this is not as comprehensive as inferring a large multiclass insect phylogeny, it was the best available method given the current state of arthropod phylogenetics. We acknowledge that this method is not without its limitations, such as the under-sampling of the list, the exclusion of many species, and using sister taxa as replacements, but it still provides valuable insights into the timing of fern herbivore diversification. Fuentes-Jacques et al. documented a total of 809 fern-feeding arthropod herbivores. While this is clearly an underestimate and indirect, it is the first ever attempt to assembly phylogenetic data on fern-feeding arthropod herbivores.

These datasets were integrated with the largest and most robust phylogenies to date of ferns[39], flowering plants[83], and ants[42]. While some of these data have been published in other formats, this is the first time they have been integrated in this way before.

## Ancestral character state estimation

To explore the tempo, mode, and timing of nectary evolution in ferns, EFNs in flowering plants, and plant associations in ants, stochastic character maps were generated using the *make.simmap* function in Phytools[85]. Using the Akaike Information Criterion (AIC), we first tested the ARD and ER model of character evolution for binary states of nectary observation in ferns and flowering plants, and plant associations in ants. The dataset on nectaries in ferns was analyzed at the species and genus level. Species-level data in ferns included all taxa observed to have nectaries coded as present and all others unobserved were coded as absent. At the genus level, the tree was pruned using the *drop.tip* function in ape[86] to keep only a single branch per genus. Nectaries were assumed present in a genus if a single species was observed to have nectaries. Stochastic character maps in flowering plants and ants were only assessed at the species level. Based on AIC we used the best-fitting model for each ancestral character state estimation, we reconstructed the macroevolutionary history of nectaries in ferns, EFNs in angiosperms, and plant association in ants. The root prior was not specified and allowed to be internally interpreted for each analysis. We also used the empirical most likely value for the transition matrix for all simulations. In addition to estimating the probability of character states at nodes, stochastic character mapping samples character histories in direct proportion to their posterior probability and estimates changes along the branches[89]. This results in a population of possible evolutionary histories (realizations). We simulated 1000 realizations for each reconstruction. We summarized our results using the *describe.simmap* function in Phytools. For ease of visualization, we subset the data to 100 randomly selected realizations and visualized the posterior probability density of binary states from

our stochastic maps using the *densityMap* function in Phytools[85]. We used character maps to quantify the number of transitions between states by summarizing the number of transitions across each character's history.

## Lineages through time

We used the LTT function from the Phytools package in R to estimate the number of lineages through time across each dataset (*i.e.*, angiosperms with EFNs, ferns with nectaries, plant-associated ants, and fern arthropod herbivores). The classic method of estimating lineages through time takes a standard time-calibrated phylogeny and estimates the number of new lineages through time. We instead use a new method which takes into account stochastic character maps and the simulated realizations of our traits of interest[90]. For angiosperms, ferns, and ants, we randomly sampled 100 of the 1000 stochastic character maps from our previous analyses using the *sample* function in base R[91]. Using the *ltt* function in Phytools on our subset of 100 stochastic character maps we estimated lineages through time. The output shows the cumulative number of lineages in each particular character state over time[90]. Given that these are averages representing transitions the line can decrease in some spots, unlike a traditional LTT plot. The plot allows us to visually inspect the pattern of lineage accumulation in the clades and to identify potential shifts in the patterns of lineage accumulation. The accumulation of lineages through time is the summation of all taxa across the entire phylogeny having the trait of interest. In order to statistically test the relationship between diversification patterns of each lineage we followed Moreau et al.[55], and employed tests of constant and time-dependent diversification implemented using the *diversi.time* function in ape. This function fits survival models to branching times. We tested three models (constant, increasing diversification through time, and a lagged diversification at a specified rate shifts across each clade).

## Timing of major events

We assessed the timing of key events including nectary origin in ferns, the subsequent Cenozoic diversification of nectary-bearing ferns, nectary origin in flowering plants, and the origin of plant associations in ants. To accomplish this, we extracted information from the LTT multi-simmaps including the timing of origin for each major event (i.e., nectary origin in ferns and flowering plants, initiation of plant associations in ants). First, we extracted all the LTT plots filtering to include only the 'gains' of the trait of interest (e.g., nectary presence, plant association) across the 100 simulations. We then filtered the data to include only nodes associated with the timing of estimated trait gains between 1 and 2 gains. We specifically selected gains between 1–2 to ensure certainty around the estimation of origin. Once data were filtered, we calculated the median timing of gains for each iteration; this number was variable for each iteration given that LTT plots can increase and decrease as they are generated from sets of stochastic character maps[90]. We then calculated the minimum, maximum, median, mean, and standard deviation of estimated timing using the median value for each of the 100 LTT simulations. To statistically compare the timing of nectary origin across ferns, flowering plants, and the origin of plant associations in ants we conducted a Wilcox test. It is important to note that the distribution of times for trait gains are conditional on the parameter values estimated from the 100 stochastic character map simulations. The timing of 'origin' denotes the earliest estimated gain of the trait of interest across the whole lineage, not the individual timing of origin across each clade where the trait evolved.

## Correlation between habitat and nectary presence

Given that the overwhelming majority of nectary-bearing ferns are canopy-dwelling, we compared the phylogenetic correlation between growth habit and nectary presence. The classic model used to explore the relationship between discrete character evolution is explained in

Pagel 1994[92]. However, this model has its limitations in certain phylogenetic scenarios where particular traits have few instances of replicated evolution[93]. To account for these issues, a new model of correlated evolution which accounts for hidden rates was developed and implemented in the CorHMM package[94]. We used the *fitCorrelationTest* function in the CorHMM package to test the phylogenetic correlation between of growth habit and nectary presence. We coded growth habit following two schemes. In the first scheme, growth habit was coded as a binary character assuming that species are either understory or canopy-dwelling. We used the term canopy-dwelling to articulate plants that elevate their leaves into the canopy or sub-canopy are because whether they are epiphytic, climbing, or tree ferns, they are elevated above the understory and either in the angiosperm canopy or creating an analogous canopy themselves. In the second scheme, we divided canopy-dwelling in three additional states (epiphyte, climber, and tree). For each coding scheme, we tested four models: a classic correlated model (*i.e.*, Pagel 1994), a character-independent model, a character-independent model assuming two rate categories independent of the focal character, and a dependent model assuming two rate categories independent of the focal character and correlation between the characters. We assessed the relative fitness of each model using corrected AIC. To visualize the data, we used stochastic character maps to simulate 100 character histories of the dual character state with hidden rates. We then used a modified version of the *densityMap* function in Phytools to plot a tree with the posterior density for the dual character states mapped from stochastic character mapping. We then used a similar approach to plot the rate shifts across the tree. We then summarized the total number of transitions across all sampled character histories using the *countSimmap* function in Phytools (Supplementary Data 2).

### Diversification analyses

We used two separate analyses to explore the correlation between nectary evolution and lineage diversification. Initially, we implemented a character state speciation extinction (SSE) mode to examine the potential impact of nectary presence on lineage diversification. Specifically, we first used BiSSE to evaluate the influence of a binary trait on the rate of diversification, excluding the potential for a hidden rate category[95]. For our analysis, we conducted 10,000 BiSSE MCMC generations using the diversitree software[96]. We employed a model with all parameters freely estimated after setting priors based on Maximum Likelihood estimates. Given the high error rates generated from standard BiSSE models and the simplistic assumption of forced differences in diversification between states[97], we used a hidden state-dependent speciation extinct model (HiSSE) to test four additional models which include hidden states. The HiSSE model addresses the issues of BiSSE by assuming a hypothetical 'hidden state' or 'hidden rate' which allows for rate heterogeneity across the tree and between character states[98]. Using the *hisse* function in the hisse package[98], we tested a total of five models: a standard BiSSE model (not distinct from the original BiSSE analysis), a null BiSSE model, a character-independent diversification model with two (CID-2) or four (CID-4) hidden states, and a full HiSSE model with two hidden states. In these models, the net turnover rates (speciation rate plus extinction rate) and eps rates (extinction fraction) were allowed to vary according to the state. We also constrained the root node to be in state nectary absence. We compared the goodness of fit of each model using AICc (Supplementary Data 3).

To further explore the relationship between nectary evolution and lineage diversification across ferns we implemented an untraditional lag-time diversification analysis, as implemented by Landis et al.[99]. The reason why we tested this unconventional approach is because it assumes that there may be a lag in diversification after the origin of a particular trait, instead of assuming a direct correlation between state shifts and rate shifts. In short, we first ran a MEDUSA

analysis on both the complete fern phylogeny and the genus-level fern phylogeny. To perform these analyses, we utilized turboMEDUSA version 0.951[100]. At the species level this analysis identified rate shifts across the phylogeny. Since the fern tree did not have 100% taxonomic sampling for all species, it was necessary to estimate species richness for known clades in the MEDUSA analyses at the genus level. To determine the number of species not represented in the tree, we gathered information from PPG I[101]; (Supplementary Data 5). To examine potential changes in diversification rates over time and their relationship with nectary gain, we employed a four-node approach. Initially, we used the *getDescendants* function in Phytools to identify nodes descended up to four steps from a nectary gain event. Subsequently, we compared these identified descendant nodes with the diversification shifts identified through the MEDUSA analysis. To examine the relationship between the diversification of nectary-bearing ferns and diversification rate shifts across fern herbivores, we explored the patterns of fern herbivore diversification and the diversification of nectary-bearing ferns. Using the fern herbivore tree generated from TimeTree5, we used Medusa to compute rate shifts (changes in diversification rates) across the phylogeny. To visualize the temporal distribution of rate shifts, we mapped each rate shift to its corresponding nodal time. We then create a sliding window analysis, with different window sizes (10, 20, 30, 40, 50, 60, 70, 80, 90, 100) and step sizes (5, 10, 15, 20, 25, 30, 35, 40, 45, 50), to partition the rate shifts into temporal bins. Each bin contains the number of rate shifts that occurred within that time window. To create a robust summary of the rate shifts, we bootstrapped this process, resampling the rate shifts with replacement and counting the number of resampled rate shifts in each time window. We repeat this process for different window sizes, creating a collection of data frames representing the distribution of rate shifts over time for each window size. These rate shift data and LTT plots of ferns with nectaries were then combined into a single plot to provide a comprehensive view of the temporal distribution of rate shifts and the diversification of ferns with nectaries.

### Reporting summary
Further information on research design is available in the Nature Portfolio Reporting Summary linked to this article.

## Data availability
The entire data generated in this study have been deposited in the GitHub database under accession code https://sandbox.zenodo.org/records/49427 (https://github.com/Suissajacob/FernNectaries).

## Code availability
The R code for the current study is publicly available with the Source Data on GitHub(https://github.com/Suissajacob/FernNectaries) https://sandbox.zenodo.org/records/49427.

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

## Acknowledgements

We thank our two undergraduate research technicians, Emileen Flores and Sydney Colón for assistance in data collection. We also thank David Barrington, Robin Moran, Andrew Murdock, Shi-Yong Dong, Alejandra Vasco, Susan Fawcett, Fernando Matos, Richard White, Melvin Turner, Jose Palacios, Christian Adolfo López, and Nicole Sebesta for expert correspondence and assistance in coding nectary presence in ferns. We also thank Jacob Landis for providing code for Medusa analyses, and to James Boyko and Liam Revell for helpful discourse on phylogenetic comparative methods. We also thank Sylvia Kinosian, William Friedman, Daniel Faccini, Faye Rosin, James Fortin, and Júlia García Güell for helpful comments on the manuscript. This work was funded in part by The National Science Foundation Award ID: 2210800 awarded to J.S.S. and Award ID: 2210800 awarded to C.S.M. Partial funding for open access to this research was provided by University of Tennessee's Open Publishing Support Fund.

## Author contributions

Conceptualization: J.S.S., F.W.L., C.S.M. Methodology: J.S.S. Investigation: J.S.S. Writing – original draft: J.S.S. Writing – review & editing: J.S.S., F.W.L., C.S.M.

## Competing interests

The authors declare no competing interests.
