## [Peer Review File · Nature Communications]

Convergent evolution of fern nectaries facilitated independent recruitment of ant-bodyguard from flowering plantsReviewers' Comments:

Reviewer #1:

Remarks to the Author:

Suissa et al. presents a phylogenetic comparative analysis of extrafloral nectaries in ferns. The key conclusions are that EFNs evolved relatively late in ferns, most of which following angiosperms, and the authors attribute this delayed to the rise of fern-feeding arthropods in the Cenozoic.

A study of EFN evolution in ferns is certainly welcome and would make an important contribution to the literature on indirect defense and mutualisms in general, but the paper is impaired by a number of significant flaws, which are addressable but require substantial re-running of analyses and re-writing of the manuscript.

Here are my concerns:

1) My most important concern is the way the authors pulled together epiphytes, climbers and tree ferns as 'arboreal'. This is problematic at various levels. First, climbers and tree ferns are terrestrial, so conflating them is simply flawed. Second, it seems very likely that trait changes between climbing <-> non-climbing, non-tree fern terrestrial, tree fern <-> non-climbing, non-tree fern terrestrial and epiphyte <-> non-climbing, non-tree fern terrestrial are different, and hence merging them into the same bin clearly bias the analysis. Going back to the materials and Methods I'm seeing both coding schemes appear to have been run, but this should be more clearly explained and showed in the figures, since the grouping 'arboreal' bias the results – at least intuitively—towards your expectation.

2) L. 317-320: 'While we uncover shifts in fern herbivore diversification rates and an accumulation of lineages corresponding with the rise of nectary-bearing ferns (Fig. 4; S7, S8, S10), these patterns are not statistically supported (Table S3).'

The interpretation of the delayed diversification of EFN-bearing ferns matching fern-feeding arthropods is not significant. Despite this, the finding is sold from the abstract onwards, and this is misleading.

3) I have many issues with some of the wording used in this manuscript. Here are a few examples:

-Secondary defense: what you mean is indirect defense, not secondary. Primary/secondary refers to a different framework which is not relevant here. Please change this throughout the manuscript.

-The manuscript is loaded with flowery terms that are misleading, eg. 'the coevolution of complex mutualism' referring to EFN-based indirect defense. First there is no evidence that EFN-bearing plants and EFN-feeding ant guilds are coevolved (and I think it is always wise to go back to Janzen's 1980's short paper 'When is it coevolution' in this regard). Second, what do you mean exactly by 'complex mutualism'? It can be viewed as complex from a network point of view as you have multispecies guilds in both sides, but otherwise, what you have is generalist and facultative, fairly simple mutualism. Simpler words would help the reader.

-L. 88: please provide a better definition for domatia, rather than just the Greek translation. E.g. a derived plant-made structure where symbiotic ants nest.

-L. 88: Beltian bodies are not starch deposits. They are protein and lipid rich. Just read the fabulous work of Martin Heil and colleagues.

-L. 91 'by attracting pugnacious ants'. Remove pugnacious. Many ants that feed on EFNs are not aggressive at all. And in fact the efficiency of this mutualism has often been questioned. There is whole body of literature when and why EFNs may or may not be efficient.

L. 207-208 'ancient symbioses'. (and L. 342). No matter how old these interactions, EFN-bearing plants/ ant mutualisms are not symbioses. Symbiosis was coined by Anton de Bary in 1879 to refer to 'the living together of unlike organisms'. The history of the term, and why mutualisms and symbiosis are often mistakenly equated is actually very interesting (cf. Bronstein 2015 Mutualism book). Some scholars, e.g. Angela Douglas, define symbioses as 'intimate mutualisms', but even in this broader cases, your interactions don't fall.

L. 355 (and elsewhere) 'ferns independently co-opted an army of ant defenders by tapping into established ant-angiosperm interactions'. This sort of statement oversells your story. Please remove 'army', cf my comments on many of these ants not being very aggressive. The word 'co-option', which you used throughout from the title onwards, is a bit misleading. Co-option typically refers the evolutionary recycling of a trait, or a genetic pathway, and in fact you use it for this meaning L. 175-176. Co-option really refers to an evolutionary mechanism. In your case here, there is no evolutionary mechanism involved at all. EFN-bearing angiosperms predated EFN-bearing ferns, and but ants are not co-opted by ferns ! At best, they are recruited. I would strongly suggest the authors to change this throughout the ms.

Along a similar vein, a cohort of journalistic-style vocabulary should be changed to more descriptive scientific words. Examples include 'innumerable', 'ant-bribing', 'intriguing', 'myriad' etc.

Reviewer #2:

Remarks to the Author:

Congratulations on the incredibly inspiring and exciting manuscript!

The authors shed light on some intriguing results regarding the evolution of nectaries in ferns and flowering plants. As outlined in the abstract, the key findings are summarized below:

- 1- Nectaries in ferns and flowering plants emerged simultaneously during the Cretaceous period, coinciding with the evolution of plant associations with ants;
- 2- While nectaries in flowering plants exhibited a steady evolutionary progression over time, those in ferns displayed a significant delay of nearly 100 million years between their origin and subsequent diversification in the Cenozoic;
- 3- The diversification of nectaries in ferns coincided with the emergence of fern-feeding arthropod herbivores during the Cenozoic;
- 4 - There appears to be a correlation between the evolution of nectaries and shifts in growth habits, suggesting that ferns utilized ant bodyguards from pre-existing ant-angiosperm relationships during their transition from the forest floor to the canopy.

The manuscript is original, robust, and of significant value, particularly for studies concerning the evolution of nectaries in plants and the coevolution of plants and ants. The study was meticulously designed and employed robust methodologies. The utilized databases appear highly reliable. The supplementary dataset and supporting information are comprehensive and well-organized, facilitating method reproducibility. The analyses appear thorough, consistently striving for maximal support for the study's results and conclusions. Thus, the findings and conclusions of the work appear strongly supported.

Reviewer #3:

Remarks to the Author:

I enjoyed reading this manuscript. It is a solid piece of work showcasing the timeline of nectary

evolution in ferns and angiosperms together with the rise of ants and to some degree perhaps also with the evolution of fern-feeding insects. The manuscript is based on a meta-analysis of four datasets that were mainly drawn from the literature. For that reason, its main contribution is the analysis of the possible evolutionary correlation between nectary evolution in plants, fern-feeding insects, and ants. The authors conclude that nectaries of ferns and EFNs in angiosperms evolved parallel and together with the rise of ants, but that nectary-bearing fern lineages diversified 100 my later than in angiosperms.

Results are well presented and discussed, except for a few details that are erroneous or some relevant gaps that should be discussed in more detail. Although meta-analyses are a fantastic tool to understand the big picture, they can miss important biological details, for instance by coding each nectary, fern-feeding insect or ant species equally, even if they play very different roles. For instance, ants are not the only insects feeding on nectaries, and many ant species do not provide any plant protection in response to nectar feeding. In addition, some sap-sucking insects are herded and protected by ants, so that plant nectar production has to compete with insect "nectar"! Consequently, the rise of these plant-feeding insect groups (e.g., Coccoomorpha, in the Triassic, Veà and Grimaldi 2016, doi: 10.1038/srep23487, or Fulgoromorpha in the Cretaceous, Bucher et al. 2023, doi: 10.1016/j.ympcv.2023.107862) might have been much more important than other herbivorous insects groups. This competition between plant nectaries and insect "nectar" to attract ants should be discussed because it might have played a role in the nectary evolution. When comparing fern and angiosperm EFNs, another aspect should be discussed briefly. Ants can be attracted by the floral nectar of angiosperms, while this competition between two kinds of nectar is missing in ferns.

Detailed comments:

L161-164: ferns and seed plants have a common ancestor, not ferns and angiosperms! Angiosperms and gymnosperms split later into two lineages. (also on L237-238)

L164: The cited reference of Testo and Sundue (2016) gives a crown age of 431 mya for ferns. Why do you use 375 mya?

L166-168: The origin of nectaries at the BASE of three families is indeed intriguing. Why would such a supposedly useful trait develop under evolutionary pressure and then get lost by 95% of their plant species?

What is the nectary origin in the other two families (see L213)?

L178-180: Sorry, I did not catch this idea. What has driven plants to evolve nectaries if ants feed on wound sap? This kind of ant-feeding is neither beneficial nor antagonistic (nor mutualistic) but rather accidental and occasional, so it should not exert any selective pressure on plants to produce nectaries (also L 252-254). Please clarify.

L228-229: The variable position in *Drynaria* has been reported earlier, see Janssen and Schneider (2005) doi: 10.1007/s00606-004-0264-6

In addition, ref 17 showed that nectaries may release nectar to the lower or upper leaf surface.

L269-273: Ambiguous statement. Please rewrite, for instance: "disproportionately enriched in non-terrestrial lineages but also including the terrestrial tree ferns *Cyatheaceae*...". In the following sentence, it becomes clear that you want to say that these three growth forms develop their leaves in the canopy (or subcanopy).

L273-274: This idea is interesting (!), but I would not call climbers or epiphytes "arboreal", this is an easily misleading term (for instance on L 281-282). Perhaps rewrite: Since these lineages mostly develop their leaves in the (sub)canopy we will group them into "canopy-dwellers" (versus "understory ferns").

L276-277: Could this be simply the consequence of 90% of new fern lineages being epiphytic? (as you say on L322-324).

L329-344: I agree with the authors but wonder why they do not consider that their results may prove that extrafloral nectaries just do not provide sufficient benefit to ferns to be of selective value. Most ferns produce nectar at low rates, only on young, developing leaves, and with low sugar content (ref. 17, 22, 26), so they are not continuous nectar providers. The SAME nectary-bearing ferns surely have other herbivore defenses as well and are not necessarily limited to ant defense. Many ants feed on the

nectar as opportunists without providing any protection.

L350: better say "extrafloral nectary evolution" (to exclude floral nectaries)

L356-358: This statement is fine but contradicts L161-164. The reason for this contradiction is that ferns and seed plants share a common ancestor, not ferns and angiosperms.

Fig. 2: Although the picture in the center is nice, it covers the main branches of the phylogenetic tree. The Pteridiineae are graphed in an awkward position because they belong to older branches (also in Fig. 5.). They should be near Dennstaedtiineae and Lindsaeineae (between 5 and 6 o'clock). If the root is at 3 o'clock the Polypodiineae should reach this position as the most advanced taxon.

How do you know that the old group of Schizaeales did not evolve nectaries earlier, but those groups have gone extinct?

Fig. 4. Unclear: Which is the larger line? What color is it?

In the figure legend, the last sentence is incomplete.

Fürstenberg-Hägg et al. 2013 (doi: 10.3390/ijms140510242) reports the start of nectar-feeding around 200 mya. Why is there such a great difference compared to your figure?

Kind regards,
Klaus Mehlreter

Response to Reviewer Comments

Reviewer #1 (Remarks to the Author):

Suissa et al. presents a phylogenetic comparative analysis of extrafloral nectaries in ferns. The key conclusions are that EFNs evolved relatively late in ferns, most of which following angiosperms, and the authors attribute this delayed to the rise of fern-feeding arthropods in the Cenozoic.

A study of EFN evolution in ferns is certainly welcome and would make an important contribution to the literature on indirect defense and mutualisms in general, but the paper is impaired by a number of significant flaws, which are addressable but require substantial re-running of analyses and re-writing of the manuscript.

Here are my concerns:

1) My most important concern is the way the authors pulled together epiphytes, climbers and tree ferns as ‘arboreal’. This is problematic at various levels. First, climbers and tree ferns are terrestrial, so conflating them is simply flawed. Second, it seems very likely that trait changes between climbing <-> non-climbing, non-tree fern terrestrial, tree fern <-> non-climbing, non-tree fern terrestrial and epiphyte <-> non-climbing, non-tree fern terrestrial are different, and hence merging them into the same bin clearly bias the analysis. Going back to the materials and Methods I’m seeing both coding schemes appear to have been run, but this should be more clearly explained and showed in the figures, since the grouping ‘arboreal’ bias the results – at least intuitively—towards your expectation.

We understand the reviewers concern with our coding scheme. As the reviewer has pointed out we have also analyzed the data to include epiphytes, climbers, terrestrial, and arboreal species as different coding schemes. We have explained this in the Supporting Information see lines 100–115 and Table S1.

We want to highlight that reviewer 3 suggested that this coding scheme of “arboreal” vs. “terrestrial” is more of a semantic problem than a methodological one. They suggested that the more appropriate trait of interest is that each of these lineages (epiphytic, tree, and climbing ferns) have figured out how to elevate their leaves off the forest floor. We agree with both reviewers that “arboreal” vs “terrestrial” is not a good set of terms to use because they clearly articulate different traits than what we are interested in. Rather, reviewer 3 suggested we change the name of these traits to “canopy” vs. “understory” in order to more accurately reflect our trait of interest. The adoption of this revised coding scheme increases the accuracy of our trait characterization and provides stronger justification for aggregating climbers, tree ferns, and epiphytes. This modification is significant because it underscores our interest in exploring the relationship between nectary presence and the ability of fern species to elevate their leaves into the canopy (given the predominance of arboreal nesting among ants; Nelsen et al., 2023 *Evol. Lett.*).

Please see lines 102-115 of the supporting information:

“When analyzing growth habit data as a multistate character (terrestrial, epiphytic, climbing, and tree-habit), we found that a model assuming a hidden state and a lack of correlation between nectary presence and growth habit is supported (AICc: 827.66; Table S1). Given our specific question regarding how ferns elevate their leaves off the forest floor, we carefully considered the appropriate coding of growth habit data. We decided to treat growth habit as a binary character, distinguishing between understory and terrestrial habitats based on whether species were in the understory (non-arboreal terrestrial species, coded as understory-dwellers) or elevate their leaves off the forest floor (epiphytic, climbing, or arboreal, all coded as canopy-dwellers). Treating growth habit in this way is more appropriate given our specific question because we are interested

in how ferns elevate their leaves off the forest floor. By categorizing growth habit as understory versus terrestrial, we emphasize the distinction between ferns that primarily grow close to the forest floor and those that have adapted to elevate their leaves off the ground. When treating growth habit as a binary character (understory vs. terrestrial) we found that a model assuming the correlated evolution of nectary presence and growth habit with a hidden rate best fit the data (AICc: 750.75; Fig. 5a–d; Table S1).”

2) L. 317-320: ‘While we uncover shifts in fern herbivore diversification rates and an accumulation of lineages corresponding with the rise of nectary-bearing ferns (Fig. 4; S7, S8, S10), these patterns are not statistically supported (Table S3).’ The interpretation of the delayed diversification of EFN-bearing ferns matching fern-feeding arthropods is not significant. Despite this, the finding is sold from the abstract onwards, and this is misleading.

We toned this down to make it clear that this is not supported by our data, but could potential be a hypothesis. We removed this from the Abstract and the Conclusions which we think helps. We have also modified our text in the introduction on line 127-130 to caution this hypothesis:

“However, the diversification of ferns with nectaries shows a nearly 100My delay between from their origin and subsequent diversification; this lag possibly may relate to the rise of fern-feeding arthropod herbivores in the Cenozoic, although the evidence supporting this hypothesis is presently limited.”

3) I have many issues with some of the wording used in this manuscript. Here are a few examples:

-Secondary defense: what you mean is indirect defense, not secondary. Primary/secondary refers to a different framework which is not relevant here. Please change this throughout the manuscript.

We have changed the language from “secondary defense” to “indirect defense” in all instances in the manuscript.

-The manuscript is loaded with flowery terms that are misleading, eg. ‘the coevolution of complex mutualism’ referring to EFN-based indirect defense. First there is no evidence that EFN-bearing plants and EFN-feeding ant guilds are coevolved (and I think it is always wise to go back to Janzen’s 1980’s short paper ‘When is it coevolution’ in this regard).

We have modified the language to remove loaded terms as the reviewer suggested.

For instance, we removed complex coevolution to mutualism, see line 51-53 of the Abstract:

“Identifying the evolutionary processes underpinning these indirect defenses provide insight into the evolution of mutualism.”

We change coevolution to mutualism, see line 78-80:

“Identifying the processes underpinning these indirect defense strategies is paramount in understanding how mutualism can shape the diversity of complex traits.”

We changed coevolutionary to simply evolutionary, see lines 158-160:

“Alternatively, it could reflect an ecological or evolutionary predisposition to evolving nectaries—as hypothesized to explain the phylogenetic patterns of EFNs across the whole Fabaceae³⁸.”

We changed coevolution to mutualism in, see line 283:

“Evolutionary lags are common patterns observed in mutualism.”

Second, what do you mean exactly by ‘complex mutualism’? It can be viewed as complex from a network point of view as you have multispecies guilds on both sides, but otherwise, what you have is generalist and facultative, fairly simple mutualism. Simpler words would help the reader.

We have removed the term “complex mutualism” from the manuscript. The term “complex” itself only appears when talking about a trait in reference to the multispecies interaction, as the reviewer suggests.

-L. 88: please provide a better definition for domatia, rather than just the Greek translation. E.g. a derived plant-made structure where symbiotic ants nest.

This was modified, see lines 83-84:

“These rewards include domatia (e.g. plant-made structure where symbiotic ants or other organisms nest)...”

-L. 88: Beltian bodies are not starch deposits. They are protein and lipid rich. Just read the fabulous work of Martin Heil and colleagues.

This was modified, see lines 84-85:

“...Beltian bodies (protein or lipid-rich deposits¹³)”

We also added the citation 13 from Heil et al., 2004 (Heil, Martin, et al. “Main nutrient compounds in food bodies of Mexican Acacia ant-plants.” *Chemoecology* 14 (2004): 45-52.)

-L. 91 ‘by attracting pugnacious ants’. Remove pugnacious. Many ants that feed on EFNs are not aggressive at all. And in fact the efficiency of this mutualism has often been questioned. There is a whole body of literature when and why EFNs may or may not be efficient.

Pugnacious was removed, see line 87-88

“By attracting ants, EFNs are an effective...”

L. 207-208 ‘ancient symbioses’. (and L. 342). No matter how old these interactions, EFN-bearing plants/ant mutualisms are not symbioses. Symbiosis was coined by Anton de Bary in 1879 to refer to ‘the living together of unlike organisms’. The history of the term, and why mutualisms and symbiosis are often mistakenly equated is actually very interesting (cf. Bronstein 2015 *Mutualism* book). Some scholars, e.g. Angela Douglas, define symbioses as ‘intimate mutualisms’, but even in these broader cases, your interactions don’t fall.

We have replaced symbiosis with relationship, see lines 182–183:

“Moreover, while extant Marattiales do not have nectaries, they do have ant associations (A. G. Murdock pers. comm.), which could hint at an ancient relationship.”

L. 355 (and elsewhere) ‘ferns independently co-opted an army of ant defenders by tapping into established ant-angiosperm interactions’. This sort of statement oversells your story. Please remove ‘army’, cf. my comments on many of these ants not being very aggressive.

We have removed the word army, please see line 349-351:

“Given the lagged diversification of nectary bearing ferns and the correlation between nectary presence and shifts away from the forest floor into the canopy, it is likely that ferns independently recruited ant defenders by tapping into established ant-angiosperm interactions.”

We have also toned down the language and replaced certain words throughout the manuscript following the reviewer’s recommendation. See the following two points.

The word ‘co-option’, which you used throughout from the title onwards, is a bit misleading. Co-option typically refers the evolutionary recycling of a trait, or a genetic pathway, and in fact you use it for this meaning L. 175-176. Co-option really refers to an evolutionary mechanism. In your case here, there is no evolutionary mechanism involved at all. EFN-bearing angiosperms predated EFN-bearing ferns, and but ants are not co-opted by ferns ! At best, they are recruited. I would strongly suggest the authors to change this throughout the ms.

We have removed the word co-opt (excluding where the reviewer points out its correct use) and replaced it with recruitment throughout the manuscript including in the title as the reviewer recommended.

Along a similar vein, a cohort of journalistic-style vocabulary should be changed to more descriptive scientific words. Examples include ‘innumerable’, ‘ant-bribing’, ‘intriguing’, ‘myriad’ etc.

Following the reviewers’ recommendations we have made the following modifications:

“innumerable” replaced with “many”

“ant-bribing” replaced with “ant-attracting”

“intriguing replaced with “interesting”

“myriad” replaced with “many”

Reviewer #2 (Remarks to the Author):

Congratulations on the incredibly inspiring and exciting manuscript!

The authors shed light on some intriguing results regarding the evolution of nectaries in ferns and flowering plants. As outlined in the abstract, the key findings are summarized below:

- 1- Nectaries in ferns and flowering plants emerged simultaneously during the Cretaceous period, coinciding with the evolution of plant associations with ants;
- 2- While nectaries in flowering plants exhibited a steady evolutionary progression over time, those in ferns displayed a significant delay of nearly 100 million years between their origin and subsequent diversification in the Cenozoic;
- 3- The diversification of nectaries in ferns coincided with the emergence of fern-feeding arthropod herbivores during the Cenozoic;
- 4 - There appears to be a correlation between the evolution of nectaries and shifts in growth habits, suggesting that ferns utilized ant bodyguards from pre-existing ant-angiosperm relationships during their transition from the forest floor to the canopy.

The manuscript is original, robust, and of significant value, particularly for studies concerning the evolution of nectaries in plants and the coevolution of plants and ants. The study was meticulously designed and employed robust methodologies. The utilized databases appear highly reliable. The supplementary dataset and supporting information are comprehensive and well-organized, facilitating method reproducibility. The analyses appear thorough, consistently striving for maximal support for the study's results and conclusions. Thus, the findings and conclusions of the work appear strongly supported.

Thank you very much for the inspiring comments. We truly appreciate the positive feedback!

Reviewer #3 (Remarks to the Author):

I enjoyed reading this manuscript. It is a solid piece of work showcasing the timeline of nectary evolution in ferns and angiosperms together with the rise of ants and to some degree perhaps also with the evolution of fern-feeding insects. The manuscript is based on a meta-analysis of four datasets that were mainly drawn from the literature. For that reason, its main contribution is the analysis of the possible evolutionary correlation between nectary evolution in plants, fern-feeding insects, and ants. The authors conclude that nectaries of ferns and EFNs in angiosperms evolved parallel and together with the rise of ants, but that nectary-bearing fern lineages diversified 100 my later than in angiosperms.

Thank you very much for your positive feedback.

Results are well presented and discussed, except for a few details that are erroneous or some relevant gaps that should be discussed in more detail. Although metanalyses are a fantastic tool to understand the big picture, they can miss important biological details, for instance by coding each nectary, fern-feeding insect or ant species equally, even if they play very different roles. For instance, ants are not the only insects feeding on nectaries, and many ant species do not provide any plant protection in response to nectar feeding. In addition, some sap-sucking insects are herded and protected by ants, so that plant nectar production has to compete with insect “nectar”! Consequently, the rise of these plant-feeding insect groups (e.g., Coccoomorpha, in the Triassic, Veà and Grimaldi 2016, doi: 10.1038/srep23487, or Fulgoromorpha in the Cretaceous, Bucher et al. 2023, doi: 10.1016/j.ympcv.2023.107862) might have been much more important than other herbivorous insects groups. This competition between plant nectaries and insect “nectar” to attract ants should be discussed because it might have played a role in the nectary evolution. When comparing fern and angiosperm EFNs, another aspect should be discussed briefly. Ants can be attracted by the floral nectar of angiosperms, while this competition between two kinds of nectar is missing in ferns.

We believe these are good points and we have incorporated a few sentences on this in the discussion. See lines 245-248

“The link between flowering plants and ants may be stronger than with ferns, because EFNs may also serve a dual function by ‘distracting’ ants from visiting floral nectar—which may negatively impact reproduction^{61,62}.”

Also see lines 233-238,

“The gain of plant-associations in ants seems to slightly lag behind the origin of EFNs in flowering plants. It is possible that this may relate to the observation that ants are not the only insects to feed on extra-floral nectar. Indeed, in high elevation areas where the flowering plant genus *Inga* can grow, nectivorous ants may be absent and predatory or parasitoid wasps (lineages which predate the ants) have been observed to feed on extra floral nectar^{27,59}”

Also see lines 313-317,

“Moreover, there may be contrasting interactions between particular types of insect herbivores such as chewing and sap-sucking insects. Sap-suckers may attract ants to ‘protect’ and ‘tend’ them, leading to direct competition between insect-nectar and plant-nectar⁷⁰. A more detailed analysis of herbivore groups and the complex multi-species interactions may resolve these fine-scale relationships.”

We also want to highlight the point made in lines 80-84 of the Supporting information regarding character coding

“Overall, our results may reflect the coding of binary nectary presence, which surely simplifies the data. While sufficient detail on the development of nectaries is currently unavailable, deeper insight may be gained from examining nectary evolution as a multistate character reflecting different developmental origins (e.g., developmental similarity to hydathodes, aerenchyma (Moran, 2022), or trichomes).”

Detailed comments:

L161-164: ferns and seed plants have a common ancestor, not ferns and angiosperms! Angiosperms and gymnosperms split later into two lineages. (also on L237-238)

We have modified the sentence to more accurately reflect our point. What we meant is that ferns and flowering plants shared a common ancestor. See lines 141-144

“This is a remarkable example of convergent evolution, especially considering that ferns and flowering plants share their most recent common ancestor more than 400 million years ago^{34,35}.”

L164: The cited reference of Testo and Sundue (2016) gives a crown age of 431 mya for ferns. Why do you use 375 mya?

We were basing this off of the fossil record. However, we have modified our language to match somewhere in between the phylogenetic and fossil record. We have changed this throughout the manuscript to read “more than 400 million years”

L166-168: The origin of nectaries at the BASE of three families is indeed intriguing. Why would such a supposedly useful trait develop under evolutionary pressure and then get lost by 95% of their plant species?

This is a good question. We believe EFNs (or nectaries in ferns) are quite labile, easy to evolve and be lost. It is possible that ant-interactions may not be reliable in certain cases and losses may then be common. This is actually observed in flowering plants too, see lines 151-154.

“These observations are similar to those of EFNs in the flowering plant genus *Senna* 37, which is also hypothesized to have a single ancestral origin with many losses and positional modifications across their evolutionary history.”

What is the nectary origin in the other two families (see L213)?

See lines 144–148

“However, until now, we lacked a detailed understanding of the timing, scale, and lability of nectary evolution within each respective clade. Among ferns, we found three major ancestral

origins at the base of the Cyatheaceae, Polypodiaceae, and Lygodiaceae (we also found independent origins of nectaries in Pteridium, Dennstaedtiaceae and Polybotrya, Dryopteridaceae; Fig. 2, S1–S3).”

L178-180: Sorry, I did not catch this idea. What has driven plants to evolve nectaries if ants feed on wound sap? This kind of ant-feeding is neither beneficial nor antagonistic (nor mutualistic) but rather accidental and occasional, so it should not exert any selective pressure on plants to produce nectaries (also L 252-254). Please clarify.

This is a hypothesis proposed by primarily by Staab et al., (Staab, M., Fornoff, F., Klein, A.-M. & Blüthgen, N. *Ants at Plant Wounds: A Little-Known Trophic Interaction with Evolutionary Implications for Ant-Plant Interactions*. *Am. Nat.* 190, 442–450 (2017)), which suggests that the origin of nectaries occurred from plant wounds. It is possible that ants present on plants and feeding on plant wounds could be beneficial if the ants started to defend the plant. Over time, the individuals that started to exude sugary water would attract ants more than the individuals that did not (or only did from wounds) in this way, wounds could have been the origin of an interaction which could have driven the evolution of nectaries.

L228-229: The variable position in *Drynaria* has been reported earlier, see Janssen and Schneider (2005) doi: 10.1007/s00606-004-0264-6

In addition, ref 17 showed that nectaries may release nectar to the lower or upper leaf surface.

We have included the citation, see lines 204–207:

“Likewise, in the basket ferns *Drynaria* spp., we observed that nectaries have a single ancestral origin (Fig. 2) but are positionally variable, occurring inconspicuously near vein junctions (Fig. 1e) or on highly modified cup-like structures (Fig. 1f), also observed by others²⁵”

L269-273: Ambiguous statement. Please rewrite, for instance: “disproportionately enriched in non-terrestrial lineages but also including the terrestrial tree ferns Cyatheaceae...”. In the following sentence, it becomes clear that you want to say that these three growth forms develop their leaves in the canopy (or subcanopy). L273-274: This idea is interesting (!), but I would not call climbers or epiphytes “arboreal”, this is an easily misleading term (for instance on L 281-282). Perhaps rewrite: Since these lineages mostly develop their leaves in the (sub)canopy we will group them into “canopy-dwellers” (versus “understory ferns”).

We have modified this based on the concerns by reviewer 1 as well. We have modified “terrestrial and arboreal” to “canopy” vs. “understory.” Please see response to reviewer one for more detail.

L276-277: Could this be simply the consequence of 90% of new fern lineages being epiphytic? (as you say on L322-324).

We have added a caveat here see line 268-271.

“While it's worth noting that many nectary-bearing ferns are epiphytic⁷³, our observed patterns may not be solely attributed to epiphytism. This is underscored by the presence of nectaries in many tree ferns, suggesting that factors beyond epiphytism could also be influencing these patterns.”

L329-344: I agree with the authors but wonder why they do not consider that their results may prove that extrafloral nectaries just do not provide sufficient benefit to ferns to be of selective value. Most ferns

produce nectar at low rates, only on young, developing leaves, and with low sugar content (ref. 17, 22, 26), so they are not continuous nectar providers. The SAME nectary-bearing ferns surely have other herbivore defenses as well and are not necessarily limited to ant defense. Many ants feed on the nectar as opportunists without providing any protection.

Yes, this is possible, we have added a statement on this, see lines 309–312:

“It is possible that this observation suggests that nectaries do not provide sufficient anti-herbivore benefit to the ferns. Indeed, many ferns do have complex antiherbivore biochemistry⁶⁸, and some ants observed to feed on fern nectaries are not aggressive⁶⁹.”

L350: better say “extrafloral nectary evolution” (to exclude floral nectaries)

This edit was made.

L356-358: This statement is fine but contradicts L161-164. The reason for this contradiction is that ferns and seed plants share a common ancestor, not ferns and angiosperms.

We have modified our language previously to articulate our major point. All organisms share a common ancestor so we feel that it is evolutionarily accurate to say that ferns and flowering plants share a common ancestor at a particular timepoint.

Fig. 2: Although the picture in the center is nice, it covers the main branches of the phylogenetic tree. The Pteridiineae are graphed in an awkward position because they belong to older branches (also in Fig. 5.). They should be near Dennstaedtiineae and Lindsaeineae (between 5 and 6 o'clock). If the root is at 3 o'clock the Polypodiineae should reach this position as the most advanced taxon.

The phylogeny is in the correct orientation, the nodes are just on a swivel, and Pteridiineae is simply rotated in a different orientation, but the node arrangement is accurate based on our best estimates of the evolutionary relationships of ferns. We have modified the image transparency to allow for a better readability of the nodes.

How do you know that the old group of Schizaeales did not evolve nectaries earlier, but those groups have gone extinct?

This is possible and we highlight that in the Supporting information and main text, see lines 181-182

“This is reflected in a probability of nectary origin in Schizaeales (Fig. 2).”

Fig. 4. Unclear: Which is the larger line? What color is it?
In the figure legend, the last sentence is incomplete.

We have modified the legend to more clearly explain the graph. Please see new figure legend 4.

Fürstenberg-Hägg et al. 2013 (doi: 10.3390/ijms140510242) reports the start of nectar-feeding around 200 mya. Why is there such a great difference compared to your figure?

We were unaware of this publications, but we have incorporated it into the discussion. We were unsure how Fürstenberg-Hägg et al. calculated the 200mya age of nectar feeding in ferns. From close examination of their paper, their plot is not rooted in any data, but seems to be a broader conceptual figure. Nonetheless, we include this contradiction in timing in our manuscript, see line 167-173.

“Interestingly, our full observed time range of nectary origin in ferns encompasses previous hypotheses suggesting a Jurassic origin, around 200Mya⁴¹. However, our large dataset enables us to refine these hypotheses, pinpointing a more specific origin approximately 135Mya. This contrast with earlier hypotheses emphasizes the previous lack of supporting data, which precluded a consensus on the timing of fern nectary origin. Our comprehensive dataset provides statistical evidence, enabling us to offer a clearer understanding of nectary origin in ferns.”

Kind regards,
Klaus Mehltreter

Reviewers' Comments:

Reviewer #1:

Remarks to the Author:

My comments have all been adequately addressed. I am also happy about the way you distinguish your two groups of ferns.

Just one minor point:

L. 337: 'nectary-driven symbiosis' would need to be changed to nectar-driven mutualism.

Yours sincerely

[signed]

Guillaume Chomicki

Reviewer #3:

Remarks to the Author:

The authors have responded very well to all raised issues and integrated the corrections into their manuscript.

I have only two minor corrections to suggest, which can be easily attended.

L204-207: In *Drynaria*, many nectaries are relatively conspicuous because they lack chlorophyll and stand out as transparent (see pictures in your ref. no. 18)

L675: Literature reference no. 25>Janssen and Schneider was published in *Plant Systematics and Evolution* and should be cited under this journal name. The former journal name "Osterr. Bot Z." was abandoned in 1973.

Kind regards,

Klaus Mehltreter

Response to Reviewer Comments

Reviewer #1 (Remarks to the Author):

My comments have all been adequately addressed. I am also happy about the way you distinguish your two groups of ferns.

Just one minor point:

L. 337: 'nectary-driven symbiosis' would need to be changed to nectar-driven mutualism.

This was addressed.

Yours sincerely
[signed]

Guillaume Chomicki

Reviewer #3 (Remarks to the Author):

The authors have responded very well to all raised issues and integrated the corrections into their manuscript.

I have only two minor corrections to suggest, which can be easily attended.

L204-207: In Drynaria, many nectaries are relatively conspicuous because they lack chlorophyll and stand out as transparent (see pictures in your ref. no. 18)

This was addressed.

L675: Literature reference no. 25>Janssen and Schneider was published in Plant Systematics and Evolution and should be cited under this journal name. The former journal name "Osterr. Bot Z." was abandoned in 1973.

This was addressed.

Kind regards,
Klaus Mehltreter